# Myositis-Specific and Myositis-Associated Antibodies in Fibromyalgia Patients: A Prospective Study

**DOI:** 10.3390/biomedicines11030658

**Published:** 2023-02-22

**Authors:** Gianluca Sambataro, Martina Orlandi, Evelina Fagone, Mary Fruciano, Elisa Gili, Alessandro Libra, Stefano Palmucci, Carlo Vancheri, Lorenzo Malatino, Michele Colaci, Domenico Sambataro

**Affiliations:** 1Artroreuma s.r.l., Rheumatology Outpatient Clinic, 95030 Mascalucia, CT, Italy; 2Department of Clinical and Experimental Medicine, Regional Referral Centre for Rare Lung Disease, Policlinico “G. Rodolico-San Marco”, University of Catania, 95123 Catania, Italy; 3Department of Experimental and Clinical Medicine, Division of Rheumatology AOUC Careggi, University of Florence, 50134 Florence, Italy; 4Department of Medical Surgical Sciences and Advanced Technologies “GF Ingrassia”, University Hospital Policlinico “G. Rodolico-San Marco”, 95123 Catania, Italy; 5Internal Medicine Unit, Department of Clinical and Experimental Medicine, Division of Rheumatology, Cannizzaro Hospital, University of Catania, 95123 Catania, Italy

**Keywords:** fibromyalgia, myositis-specific antibodies, myositis-associated antibodies, Sjögren’s syndrome, antisynthetase syndrome, idiopathic inflammatory myopathy, polymyositis, dermatomyositis, myalgia, diagnosis

## Abstract

Fibromyalgia (FM) is a common rheumatologic disorder characterised by widespread muscular pain. Myalgia is also a common clinical feature in Connective Tissue Disease (CTD), and FM should be studied for the concomitant presence of a CTD. The aim of this study is to evaluate the prevalence of Myositis-Specific and Myositis-Associated Antibodies (MSA/MAA) in a cohort of FM patients. We enrolled 233 consecutive FM patients (defined according to the 2016 criteria) that did not report clinical signs of autoimmune disorders and followed them for at least one year. The patients were tested for MSA/MAA with immunoblotting. FM patients were seropositive for Antinuclear Antibodies (ANA) in 24% of cases, for MSA in 9%, and for MAA in 6%. A specific diagnosis of CTD was made in 12 patients (5.2%), namely, 5 cases of primary Sjögren’s Syndrome and 7 of Idiopathic Inflammatory Myopathy. Seropositive patients showed clinical features similar to those who were seronegative at baseline. A CTD diagnosis was associated with ANA positivity (*p* = 0.03, X^2^ 4.9), the presence of a speckled pattern (*p* = 0.02, X^2^ 5.3), positivity for MAA (*p* = 0.004, X^2^ 8.1), and MSA (*p* = 0.003, X^2^ 9.2). In conclusion, a non-negligible proportion of FM patients may be seropositive for MSA/MAA, and that seropositivity might suggest a diagnosis of CTD.

## 1. Introduction

Fibromyalgia (FM) is a rheumatic disorder characterised by widespread musculoskeletal pain, fatigue, muscle stiffness, and hyperalgesia. Moreover, it is frequently accompanied by other systemic symptoms including sleep difficulties, memory impairment (known as “fibro-fog”), and mood disorders (e.g., anxiety and depression) [1]. Irritable bowel syndrome and digestive alterations are also reported. FM is very common, with a prevalence ranging from 1 to 8% depending on the set of criteria used for disease definition [2].

Although the condition’s pathophysiology is not completely understood, FM is usually defined by a dysfunction of neurocircuits involved in the conduction of nociceptive stimuli. Hypotheses are mainly oriented towards a possible dysfunction of the central nervous system: an alteration in serotonergic-noradrenergic activity, a reduction of the binding capacity of central opioid receptors, an increased level of glutamate in the cerebrospinal fluid with a reduction of gamma-aminobutyric acid, and continuous stimulation of nociceptive fibres [3]. A peripheral origin (muscular or tendinous) has also been hypothesised and could be responsible for symptoms such as muscle fatigue, myalgia, and hyperalgesia in the entheses or muscle venters. 

These difficulties in framing this multifaceted disease are reflected in the various sets of criteria proposed over the last 30 years. 

The first classification criteria were proposed in 1990 by the American College of Rheumatology and mainly focused on pain at classical Tender Points (TPs), while in the successive set of criteria great weight was also given to other characteristic symptoms such as sleep disturbance and fatigue, also allowing for classification in the presence of comorbidities [4]. However, the main question in clinical practice is to distinguish FM pain from other conditions potentially associated with similar pain, such as thyroid dysfunction, osteomalacia, and chronic inflammatory disorders [5]. Distinguishing FM from myalgias due to other chronic inflammatory conditions or from enthesitis due to other reasons (e.g., in the context of spondyloarthritis) can be very challenging from a clinical point of view, and the main differences should be identified by serological and radiological exams [6,7]. Moreover, some Connective Tissue Diseases (CTDs) can initially have a clinical presentation similar to FM, and FM itself is quite common in CTD patients [7]. However, the possibility of an autoimmune cause underlying FM is under consideration. Some studies reported a similar prevalence of classical autoantibodies in FM patients and healthy controls, showing a rate of progression toward CTD similar to that reported in the general population [8]. These data, together with the absence of clinical, laboratory, and instrumental inflammation, are not consistent with a possible autoimmune genesis of FM. On the other hand, other recent studies found pain-sensitising autoantibodies in chronic primary pain including FM, and that the passive transfer of IgG or IgM antibodies from patient-donors to rodents caused symptoms very similar to those experienced by patients [9].

Previous clinical studies on FM patients were not focused on Myositis-Specific and Myositis-Associated Antibodies (MSA and MAA). These autoantibodies are useful biomarkers in detecting Idiopathic Inflammatory Myopathies (IIMs) and associated diseases [10]. IIMs are conditions characterised by potential inflammatory involvement of the muscle, skin, and lungs. Their clinical presentation can resemble FM in terms of asthenia, myalgia, and referred dysphagia, therefore, their exclusion is suggested before a possible diagnosis [5]. MSAs and MAAs were found to have high specificity but low sensitivity in detecting IIMs [11,12]. Despite limited data being available regarding the prevalence of these autoantibodies in healthy controls, current literature reports a prevalence of about 5–6% using Immunoblotting [12,13].

The aim of this study is to evaluate the prevalence of autoantibodies in general, and MSA/MAAs in particular, in FM patients, looking for possible differences between seropositive and seronegative FM patients.

## 2. Materials and Methods

This is a prospective study, conducted from September 2021 to December 2022, in the Rheumatology units at the University of Catania. Patients were enrolled consecutively and followed for at least one year. Inclusion criteria were age > 18 years, written informed consent to participate in the study, absence of exclusion criteria, and a diagnosis of FM according to the modified 2016 criteria [14]. The exclusion criteria were: previous definite CTD diagnosis, presence of at least one of these symptoms or signs at the first visit or in the clinical history: photosensitivity, sicca syndrome, Raynaud’s Phenomenon (RP), a previous diagnosis of serositis, inflammatory arthritis, significant haematological alterations (e.g., anaemia or leukopenia), interstitial lung disease, glomerulonephritis, delirium, seizure, psychosis, thrombosis, uveitis, chronic inflammatory bowel diseases, psoriasis, or other skin rashes that would raise suspicion of an autoimmune cause.

Clinical assessment was conducted by expert rheumatologists. All patients were clinically evaluated for the possible presence of a CTD, inflammatory arthritis, suggestive skin involvement (e.g., rashes, sclerodactyly, ulcers, significant telangiectasias), or other signs and symptoms mentioned above as exclusion criteria.

The same rheumatologists followed the patients after the enrolment. In case of the development of new signs/symptoms suggestive of an underlying disease other than FM, they are free to suggest exams deemed useful to reach a confident diagnosis.

All patients underwent the same laboratory assessment, including complete blood count, Aspartate and Alanine Aminotransferase, creatinine, urine test, complement fractions C3 and C4, C Reactive Protein, Erythrocyte Sedimentation Rate, Creatine Phosphokinase, Lactic Dehydrogenase, serum protein electrophoresis, vitamin D, alkaline phosphatase, Thyroid Stimulating Hormone, free triiodothyronine, and free thyroxine. Their autoimmune profile was evaluated by centralised, skilled operators. The profile included Rheumatoid Factor, Antinuclear Antibodies (ANA), Anti Citrullinated Protein Antibodies (ACPA), anti-Double Strain DNA, Extractable Nuclear Antigen (ENA) specificities including Anti-Ro60Kd, Anti-La, Anti-Sm, Anti-Jo1, Anti-Scl70, Anticentromere Antibodies (ACA), Anti-RNP, MSA, MAA, and Anti Neutrophil Cytoplasmic Antibodies (ANCA).

All of the samples were stored at −20 °C since the test. ANCA were evaluated by ELISA, ANA were evaluated in Indirect Immunofluorescence Assay (IFA) with HEp-2 cells, according to the instruction of the manufacturer (BION Enterprises, Des Plaines, IL, USA, cod. GUDIDB110ANK600). Microscope slides were coated with HEp-2 cells, and the positivity was visualised by adding a fluorescent (FITC) tagged anti-human antibody. The description of the distinct fluorescence patterns was made according to the latest guideline [15] using an epifluorescence microscope (Nikon Eclipse E400 Microscope) at 400× magnification. A title of ANA ≥ 1:80 was considered positive. ENA were evaluated by EIA ENA profile plus (TestLine Clinical Diagnostics Ltd., Brno, Czech Republic, code ENAp12). According to the manufacturer’s instructions, after defrosting, the samples were diluted at 1:101 and incubated at 37 °C for 30 min in antigenic-coated microtitre wells. The assay included calibrators and positive and negative controls. The wells were washed 5 times and the procedure was repeated after the Conjugate and Substrate (TMB-complete) incubation for 15 min at 37 °C. The specific antibodies were detected by colour reaction and they were read at 450 nm by a microplate reader (xMark Microplate Absorbance Spectrophotometer/Bio-Rad Laboratories, Inc, Hercules, California, USA). The data analysis was performed using Microplate Manager 6 Software (Bio-Rad Laboratories, Inc.). MSA and MAA were tested by line blot immunoassay (LIA) (EUROLINE Profile Autoimmune Inflammatory Myopathies 16 Ag (IgG)- EUROIMMUN AG-PerkinElmer company, Lübeck, Germany code DL1530-1601-3G). According to the instructions provided by the manufacturer, LIAs were performed by treating the samples with a universal solution and incubating them for 30 min at room temperature with the strips. The strips were washed 3 times for 5 min and bound autoantibodies were made visible by using HRP-conjugated secondary antibodies (incubation for 30 min) and an LIA-specific substrate (incubation for 15 min). The intensities of the LIA strips were semi-quantitatively analysed using the EUROLineScan system (Euroimmun). Borderline autoantibodies were considered negative (Appendix A).

The MSAs studied were anti-Mi2, anti-Tif1γ, anti-MDA5, Anti-NXP2, anti-SRP, anti-SAE, anti-Jo1, anti-PL7, anti-PL12, anti-OJ, and anti-EJ, while the MAAs studied were anti-Pm/scl, anti-Ro52kD, anti-Ku, and anti-RNP.

The FM diagnosis was made according to the modified 2016 criteria [14], which proved to have better accuracy in the recognition of FM patients than the most recent set proposed [16], however, Tender Points (TPs) were also evaluated manually by expert operators. TPs were evaluated by applying a strength of about 4 kg to the classic points suggested in the 1990 FM criteria [17], and considered positive in cases of pain plus grimace, flinch, or withdrawal. The diagnoses of CTD were based on specific, validated criteria [18,19,20,21,22,23,24]. Patients were considered affected by IIM with a minimum score of 5.5 (probable IIM) [22]. Diagnoses of the comorbidities associated with FM often required further radiological or histological exams. These exams were performed when deemed useful by the physicians.

Before statistical evaluation, we performed a D’Agostino-Pearson test in order to evaluate the distribution of the data. Continuous variables with a normal distribution are presented in mean ± Standard Deviation (SD), otherwise in median (1–3 Interquartile Range, IQR). Dichotomous variables are reported in proportion. Continuous variables were evaluated by the Mann–Whitney U Test, and dichotomous variables with the Chi-square test. We considered a statistical significance of *p* < 0.05.

## 3. Results

A total of 233 patients were enrolled in the study, 89.3% were female, with a mean age of 57 ± 14 years. The mean Widespread Pain Index (WPI) was 14 (±3.1), the Symptom Severity Score (SSS) was 7 ± 1.1, and the TPs count was 15 ± 1.9.

WPI, SSS, and TPs were significantly higher in female patients, which also showed a significantly higher proportion of hyperthyroidism at the baseline. The general characteristics, comorbidities, and gender differences are reported in Table 1.

### 3.1. Autoimmune Assessment

FM patients showed ANA positivity in 56 (24%) patients, of which 21 (9%) were at a titre of 1:80, 28 (12%) at 1:160, and 7 (3%) at a titre of 1:320 or greater. The pattern of ANA positivity was speckled in 73.2%, homogeneous in 3.6%, nucleolar in 5.4%, and cytoplasmic in 19.6%.

Regarding the autoantibody profile, 7 patients (3%) showed positivity for RF at a minimum titre of 2 times over the upper limit. Positivity for Anti-Ro60kD was found in 1.7%, whereas a single patient (0.4%) showed positivity for Anti-La (in this case associated with anti-Ro60kD and anti-Ro52kD), anti-DsDNA, or anti-PR3-ANCA. No other autoantibody from the ENA profile was found.

Positivity for MSA/MAA was found in 33 (14.1%) patients, for MSA in 21 (9%), and for MAA in 14 (6%). Of the 33 patients seropositive for MSA/MAA, 7 (21.2%) showed combined positivity (in 4 cases multiple positivities for MSAs). The specific positivity is reported in Table 2.

Regarding gender differences, male patients were positive for ANA in 29.2%, for MSA in 4.2%, and for MAA in 12.5%. The proportion of female patients was 23% for ANA, 9.6% for MSA, and 5.3% for MAA. The difference was not statistically significant (ANA *p* = 0.049, MSA *p* = 0.38, MAA *p* = 0.15).

### 3.2. Features of Seropositive FM Patients

FM patients seropositive for MSA/MAA were female in 90.9% of cases, with a mean age of 58 years (±14.1). Their clinical assessment found WPI 13.2 (±2.2), SSS 7.5 (±0.6), and TP 14.9 (±1.5). All of these parameters were similar compared with seronegative FM patients. 

ANA positivity was associated with the presence of MAA (X^2^ 15.9, *p* = <0.0001), but not with positivity for MSA (*p* = 0.34). Increased levels of muscle enzymes (CPK, LDH, transaminases) were not associated with autoantibody positivity for either MSA (*p* = 0.44) or MAA (*p* = 0.24).

### 3.3. Diagnoses in FM Patients

FM was the only possible cause of chronic widespread pain in the vast majority of patients who were clinically defined as FM, however, 30 patients (12.9%) also displayed other conditions that could explain their clinical picture. 

In detail, non-autoimmune conditions were reported in 18 patients. Four patients were diagnosed with axial spondyloarthritis on the basis of specific MRI features. Endocrinological disorders were found in 3 patients with osteomalacia, and 9 patients with thyroid dysfunction (5 with hyperthyroidism and 4 with hypothyroidism). A single patient had a diagnosis of Transthyretin Amyloidosis that required a positive abdominal fat pat excisional biopsy, whereas another patient with persistent, increased levels of CPK was classified as having possible statin-induced myopathy. The subject experienced a clinical and serological improvement at the suspension of that treatment. 

A definite CTD was diagnosed in 12 patients (5.2%). Figure 1 shows the proportion and the specific CTD recognised.

Primary Sjögren’s Syndrome (pSS) was diagnosed in 5 patients. Of these, 1 patient was seropositive for anti-Ro52kD alone, another for anti-Ro52kD and RF, 2 patients were seropositive for anti-Ro60kD alone, and the last for RF, anti-La, anti-Ro52kD, and anti-Ro60kD. The diagnosis was made according to the specific criteria [21], finding glandular impairment through Schirmer’s Test in all of these patients and an impaired unstimulated whole saliva test in all but 1, therefore, a minor salivary gland biopsy was not required. All of these patients did not report sicca syndrome during their first visit.

Two completely seronegative patients were classified as probable Polymyositis due to the presence of objective proximal weakness of the girdles, persistent, increased levels of CPK (3 and 5 times the upper normal limit, ULN, respectively), and suggestive findings from electromyography. 

Four patients were classified as probable Idiopathic Inflammatory Myopathy (score 5.7/5.5) due to the presence of dysphagia, objective weakness of the girdles, and increased levels of CPK (at least 1.5 x ULN). Two of these patients also displayed Interstitial Lung Abnormalities (ILA), without respiratory impairment. The patients with ILA were positive for anti-PL7 and anti-Mi2, respectively. The 2 patients without ILA were positive for ANA 1:160 with a cytoplasmic pattern and for ANA 1:320 with a speckled pattern, anti-Mi2, and anti-SRP, respectively. Another patient showed positivity for anti-PL12 and anti-Mi2, with persistently increased levels of CPK (2 × ULN), ILA, and dysphagia. After one year of follow-up, the subject developed Raynaud’s Phenomenon, Heliotrope Rash, and Mechanic’s hands (probable IIM, 7.2/5.5). Therefore, at the end of the study, 7 out of 233 FM patients were classified as probable IIM. Figure 2 reports the seropositivity of definite CTD patients in the FM cohort.

Finally, another 2 patients were classified as Undifferentiated CTD, due to the onset during follow-up of inflammatory arthritis and slightly increased levels of CPK. The first patient was seropositive for ANA 1:160 with a cytoplasmic pattern and anti-Ku, and the second for ANA 1:80 with a speckled pattern, anti-DsDNA, and anti-RNP. 

The definite diagnosis of CTD was associated with ANA positivity (*p* = 0.03, X^2^ 4.9), presence of a speckled pattern (*p* = 0.02, X^2^ 5.3), positivity for MAA (*p* = 0.004, X^2^ 8.1), and MSA (*p* = 0.003, X^2^ 9.2).

## 4. Discussion

To the best of our knowledge, this is the first study aimed at evaluating the prevalence of MSA/MAA in a cohort of patients with FM. To evaluate this, we defined the presence of FM according to the 2016 criteria, in which the exclusion of other concomitant diagnoses that might explain widespread pain is not mandatory [14]. However, we selected a group of patients with a very low likelihood of having an underlying autoimmune condition, excluding those reporting a previous diagnosis of inflammatory conditions or other suggestive autoimmune features, and, above all, those included in the classification criteria for CTD. 

Despite this, seropositivity was not uncommon. The proportion of ANA positivity in our cohort was 24%, 2 times higher than that reported in the study by Kotter I et al. [8], however, our patients were, on average, ten years older. A high ANA titre was reported in 3% of our patients, similar to the proportion reported in healthy subjects [25]. The ANA positivity found in our cohort was generally, for a low titre, similar to that reported in healthy subjects of a similar age [26].

Regarding common specific autoantibodies, the most common positivity was for anti-Ro60kD, found in 4 patients, and a single positivity for anti-DsDNA and anti-La. No positivity was found for scleroderma-specific autoantibodies or anti-Sm. The proportion of these autoantibodies was similar to that described in the literature for healthy individuals [27]. However, the patient with anti-La and anti-Ro60kD, together with 2 of the 3 other patients positive for anti-Ro60kD, also displayed glandular impairment, being classified as pSS. Therefore, in these cases, seropositivity indicated a specific disease. FM in pSS patients is described in about 31% of patients [28], and it could also represent a clinical onset of pSS. In a previous study, Gau et al. found that the risk of FM patients developing pSS increased by around 2 times [29]. Moreover, seropositivity for an autoantibody associated with pSS was also found in about a third of FM patients reporting sicca syndrome [30]. All of our patients were asked about the possible presence of sicca syndrome, but none of them reported it. This should not be a surprise: a recent study reported that, despite greater ocular damage, dry eye patients with pSS complain of less discomfort than dry eye patients without pSS [31]. 

In our cohort, MSAs showed a different distribution than those reported in healthy subjects. In the large study conducted by Li et al. involving more than 25000 individuals [27], the most common MSA was Anti-Ro52kD, followed by anti-RNP and anti-Pm/scl (0.85%, 0.25%, and 0.17%). In our study, the proportion was higher for all of these autoantibodies, and the most common was anti-Pm/scl, followed by anti-RNP and anti-Ro52kD (2.1%, 1.7%, and 1.3%, respectively). The clinical significance of this difference should be further addressed: at the time of the study, we did not find any specific autoimmune conditions in those patients positive for anti-pm/scl and anti-RNP, however, all anti-Ro52kD+ patients were classified as having pSS due to their glandular impairment.

Of our FM patients, 9% were positive for MSA. The proportion is only slightly higher than reported in the literature for healthy subjects [12,13], however, it should be considered that the selection criteria used in this study were very strict, excluding patients with suggestive autoimmune features. A recent, retrospective study evaluated the prevalence of MSA in a cohort of patients with a suspicion of underlying IIM, finding positivity for MSA in 18% of patients [32]. In this cohort, as reported in our study, the most common MSA was anti-Mi2. These autoantibodies are associated with mild forms of dermatomyositis, characterised by milder myopathy, fewer occurrences of malignancy and Interstitial Lung Disease (ILD), a good response to steroids, and persistent disease remission [33]. Of the 13 anti-Mi2 + FM patients, only 3 had a specific diagnosis of IIM. These patients had skin involvement in only 1 case, all 3 patients had a mildly increased CPK level, and lung involvement was limited to ILA rather than ILD. The number of patients and the length of follow-up were not sufficient to draw definite conclusions about their prognosis. These autoantibodies could represent a current, mild form of IIM, or, in some cases, an early onset with potential for progression. Therefore, these patients require periodical follow-up. Similar observations could be made regarding the 9 patients who were positive for antisynthetase antibodies. At the time the study was closed, only 2 of these patients were classified as IIM (one positive for anti-PL7 and the second for anti-PL12). These two autoantibodies are mainly associated with ILD, and mild muscular involvement, however, a clinical onset with only myositis is not uncommon [34]. Another 3 patients were seronegative for MSA. As already reported, MSAs are useful for the diagnosis of IIM, however, they show very low sensitivity and up to 50% of IIM patients can be seronegative [35]. This proportion is quite high, and it could probably be reduced with further knowledge of the subset of autoimmunity.

Finally, no differences were found in the clinical presentation of FM between seropositive and seronegative patients. Despite a trend of lower WPI and SSS values in MSA+ patients, the difference is not statistically significant. Of course, this data should be confirmed in larger studies, however, as things stand, we cannot suggest how patients should be evaluated for MSA/MAA. The gender difference was noted in the clinical presentation of FM patients (WPI, SSS, TPs), but not in the seropositivity or the CTD diagnoses. The results of the clinical presentation are similar to what was already reported [36]. However, our study was not powered to evaluate gender differences in seropositivity, enrolling a very low number of male patients.

This study has some limitations. First of all, the study is merely descriptive, and the difference in MSA/MAA positivity in FM patients compared to controls could have been hypothesised by previous studies performed using the same techniques [12,13]. Immunoprecipitation (IP) is the current gold standard for the recognition of MSA/MAAs, however, this technique is quite difficult to perform in clinical practice. We cannot exclude some cases of false positivity, also considering that MSAs are commonly mutually exclusive [34], and, in our cohort, 4 patients (7.1% of the sera were positive for at least one autoantibody, 1.7% of the total cohort) were positive for at least 2 MSAs. In an interesting study, Cavazzana et al. found multiple seropositivities with IB in 17% of the sera evaluated, but 0% in those studied with IP [37]. Our proportion of multiple MSA positivity appears to be lower. In fact, we used the same technique as Espinosa-Ortega et al. and a similar kit, finding higher concordance with IP [13]. In this latter study, a lower concordance was noted for anti-Mi2. The authors explained this with the fact that, differently from IP, IB included both α and β proteins of the molecule. Another limitation of the study is the short follow-up time: other possible diagnoses in future years cannot be excluded, and patients are currently in follow-up for this reason.

The study also has some merits. This is the first study aimed at describing MSA/MAA positivity in a relatively large cohort of FM patients and it also evaluated other autoantibodies. The inclusion criteria were designed in order to approximate FM patients to healthy subjects, with a low likelihood of them being seropositive. It is probable that the inclusion of patients in which FM is associated with other signs (e.g., RP or skin rashes) could have increased this proportion. In fact, a recent study reported that pain similar to that reported in FM is associated with an increased likelihood of being seropositive for MSA/MAA in ILD patients [38]. Finally, a small group of FM patients actually had a CTD, suggesting the clinical relevance of periodically re-evaluating FM patients during follow-up. 

## 5. Conclusions

A non-negligible proportion of FM patients can be seropositive for autoantibodies, including MSA/MAAs. Seropositivity is not associated with different clinical features of FM at baseline. However, a diagnosis of CTD was formulated in a subset of these cases during the one-year follow-up. These findings underline the possibility of a misdiagnosed CTD in patients with a clinical picture of FM or the eventual evolution towards an overt CTD in patients diagnosed as FM at baseline.

## Figures and Tables

**Figure 1 biomedicines-11-00658-f001:**
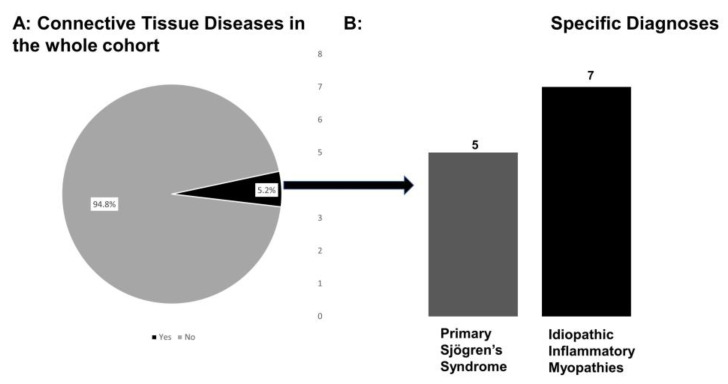
Legend: (**A**): Proportion of Connective Tissue Disease (CTD) in the whole cohort (reported in proportion); (**B**): Specific diagnoses made in the patients with an underlying or that developed a CTD in the cohort.

**Figure 2 biomedicines-11-00658-f002:**
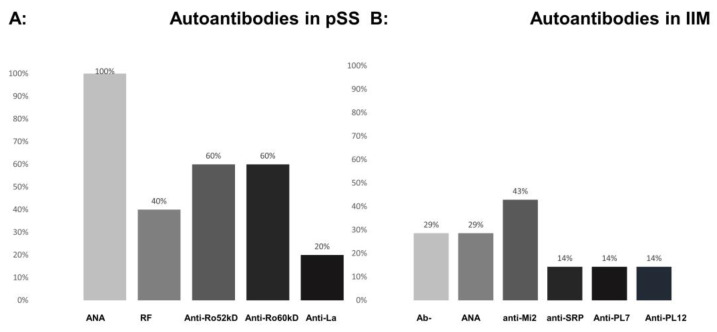
Autoantibody positivity in Fibromyalgia patients diagnosed as having a Connective Tissue Disease Legend: (**A**,**B**): seronegative patients; ANA: Antinuclear Antibodies; CTD: Connective Tissue Disease; IIM: Idiopathic Inflammatory Myopathy; pSS: primary Sjögren’s Syndrome; RF: Rheumatoid Factor.

**Table 1 biomedicines-11-00658-t001:** General features of Fibromyalgia cohort.

Item	Whole Cohort (233)	Male Patients (24)	Female Patients (209)	*p*
Age (years)	57 ± 14	54 ± 16	58 ± 13	0.19
WPI	14 ± 3.1	9.5 ± 2.5	14.5 ± 2.7	<0.0001
SSS	7 ± 1.1	6.6 ± 0.8	7.1 ± 1.1	0.03
TPs	15 ± 1.9	10.3 ± 2.5	15.6 ± 0.8%	<0.0001
Atrial Fibrillation	1.7%	0%	1.9%	0.49
Other Arrhythmias	4.3%	8.3%	3.8%	0.3
Dyslipidemia	24.5%	25%	23.6%	0.94
Systemic Hypertension	42.5%	41.7%	42.8%	0.93
T2DM	12.9%	12.5%	12.9%	0.95
GERD	15%	12.5%	15.3%	0.71
Hypothyroidism	14.2%	0%	15.8%	0.04
Depression	10.3%	12.5%	10%	0.7
Asthma	3.9%	0%	4.3%	0.3
History of MI	3.9%	0%	4.3%	0.3
History of Stroke	3%	8.3%	2.1%	0.1
History of Cancer	7.3%	8.3%	7.2%	0.83

Legend: SSS; GERD: Gastro-Oesophageal Reflux Disease; MI: Myocardial Infarction; Symptom Severity Score; T2DM: Type 2 Diabetes Mellitus; TPs: Tender Points; WPI: Widespread Pain Index. Data are reported in mean ± Standard Deviation or in proportion.

**Table 2 biomedicines-11-00658-t002:** Incidence of Myositis-Specific and Myositis-Associated Autoantibodies in the Fibromyalgia cohort.

Autoantibody	Incidence	Autoantibody	Incidence
MSA	9%	MAA	6%
Anti-Mi2	5.1%	Anti-Pm/scl	2.1%
Anti-MDA5	1.3%	Anti-RNP	1.7%
Anti-SRP	0.8%	Anti-Ro52Kd	1.3%
Anti-Tif1γ	0.8%	Anti-Ku	1.3%
Anti-PL7	1.3%		
Anti-PL12	0.8%		
Anti-Jo1	0.4%		
Anti-OJ	0.4%		

Legend: MAA: Myositis-Associated Antibodies; MAA: Myositis-Specific Antibodies; Multiple positivities for MSA/MAA were the following: Anti-SRP + Anti-Mi2; Anti-SRP+ Anti-PL7; Anti-PL12+ Anti-Mi2; Anti-PL7+ Anti-Mi2+ Anti-Tif1γ; Anti-Pm/scl+ Anti-OJ; Anti-Pm/scl+ Anti-Mi2; Anti-Pm/scl + anti-RNP.

## Data Availability

The data that support the findings are available as Appendix A.

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
