# Peer review of "Myositis-Specific and Myositis-Associated Antibodies in Fibromyalgia Patients: A Prospective Study"

_biomedicines, 2023, doi:10.3390/biomedicines11030658_

Round 1

Reviewer 1 Report

This study evaluates the prevalence of autoantibodies in general, particularly focusing on MSA/MAAs in FM patients to find any differences between seropositive and seronegative FM patients. 

The full for for ANA is not included in the abstract. Please change to Antinuclear Antibodies (ANA) on first use in line 26.

Figure 1 contains four separate graphs. I think it would be clearer to separate these. An alternative would be to label each on a - d. This helps with the description in the legend below. This legend should also be included in line 205. As it is, it appears as text in the results section.

Line 247: Change "Li X et al" to "Li et al"

Line 255: Change "9%" to Nine percent" or rephrase as " Of our FM patients, 9% were positive for MSA."

Line 266: What does ILD stand for?

Line 292: Change "Cavazzana I et al" to "Cavazzana et al"

Author Response

Thank you forthe good evaluation of our study, and your comments aimed to improve our manuscript. We accepted all of them

The term ANA was clarified (Antinuclear Antibody) in the abstract, line 26-27

We agree with your opinion for the opportunity to use another figure.  We divided the figure into two other figures. Figure 1 with its caption in line 221-226, figure 2 with its caption in line 252-259. Probably now the manuscript is more readable

"Li X et al" was changed to "Li et al", now at line 301

We rephrased line 255 as you suggested, now in line 309

The term ILD in line 266 is specified in line 316-317.

"Cavazzana I et al" was changed to "Cavazzana et al" as you suggested, now in line 350

Thank you again 

Best Regards

Reviewer 2 Report

Thank you for allowing me to review the manuscript "Myositis-Specific and Myositis-Associated Antibodies in Fibromyalgia patients: a prospective study.

The study deals with a very interesting topic, since it aims to evaluate the prevalence of Myositis-Specific and Myositis Associated Antibodies (MSA/MAA) in a cohort of FM patients.

However, before considering it for publication, the following concerns should be addressed:

1 add a table that summarizes the main data of the study

2 please assess if MSA and MAA have a sex-related difference

3 please assess any relationship between gender-related factors (age, BMI, comorbidities, pain severity…)

4. please divide figure 1 into different figures, to improve its visualization and readability 

Author Response

Thank you for your comments.

Our study is mainly directed to evaluate seropositivity for MSA/MAA in FM patients, therefore probably the main data are reported in table 2 and in figure 1 and 2. Despite our study was not powered to evaluate gender difference in FM patients (we included only 24 male patients), we agree with you: an analysis of gender differences could be useful to better describe our cohort.

We added table 1 as you suggested including difference in age, WPI, SSS, TPs, comorbidities in the two groups (male and female, table 1). Unfortunately, BMI was not systematically collected, but probably it does not significantly impact this study. Difference in seropositivity between male and female patients is reported in line 193-196. We briefly discussed the gender difference in the clinical presentation of FM patients in line 347-341.

As you suggested, we added an additional figure, in our opinion significantly improving the readability of the manuscript. Figure 1 with its caption in line 221-226, figure 2 with its caption in line 252-259

We hope that this new version could be appreciated by you.

Kind Regards

Reviewer 3 Report

Importantly approval of the study by a competent institution ethics committee seems missing in the manuscript.

Sambataro and coauthors report in their descriptive paper the detection of several autoantibodies including anti-ANA and Myositis-Specific and Myositis Associated Antibodies (MSA/MAA) and other autoantibodies in a cohort of 233 fibromyalgia (FM) patients (majorly females) showing reactivity with different cellular and nuclear patterns. The findings seem to support autoimmunity in FM patients in partial accordance with the literature. The definition of the autoantibody types is relevant towards tailored treatments. The authors correctly state in the Discussion the limitation due to the low positive numbers and invite to enlarge cohorts in continuation studies to obtain more faithful epidemiologic data. In general, this reviewer finds the paper very well written and well-structured facilitating reader´s task.

However, some points for improvement have been detected as follows:

In addition to fullfil 2016 ACR fibromyalgia clinical criteria, about 12% of patients presented varied comorbidities including CTD, SjÓ§gren’s Syndrome, IIM and other.  However, the authors indicate in the summary the cohort “did not report clinical signs of autoimmune disorders”. It would be important to detail how was this assessed at enrolment that was different in the one-year patients follow-up. This could clarify whether the analyzed cases developed secondary comorbidities to FM or if there is any data to support it was missed by clinicians at enrolment (differently assessed?).

All raw data should be made available as supplementary tables, so that readers count with more detail on individual observed values rather than only global group values. These suppl. Tables should include all analytics that are mentioned in the text, such as TH, CPK, transaminase, complement values, etc.

Since the number of seropositive cases is low, it is recommended to present the data in Figure 1 tabulated indicating the number of cases rather than percentages.

Importantly, representative images of the the autoreactive different patterns observed as described: “positivity was speckled in 73.2%, homogeneous in 3.6%, nucleolar in 5.4% and cytoplasmic in 19.6%.” should be provided to illustrate the claims. In methods provider and catalogue numbers for all antibodies used should be made available. Also, describe protocols followed to assess seropositivity, type of cells analyzed (isolation method) and the software to assess positivity and signal subcellular distribution.

Author Response

We are very grateful for your comments, we believe that it could significantly improve our manuscript

  1. As already assessed in line 383-385, the manuscript was approved by our local ethical committee, the Institutional Review Board of Catania 1 (protocol code 35836, August, 2, 2021).
  2. Thank you for your comment. As assessed in the method section, the study was conducted enrolling patients without sign/symptoms suggestive for an underlying CTD at the first visit, or in the clinical history. In particular, we excluded patients referring photosensitivity, sicca syndrome, Raynaud’s Phenomenon, a previous diagnosis of serositis, inflammatory arthritis, significant haematological alterations (e.g. anaemia or leukopenia), interstitial lung disease, glomerulonephritis, delirium, seizure, psychosis, thrombosis, uveitis, chronic inflammatory bowel diseases, psoriasis or other skin rashes that would raise suspicion of an autoimmune cause. We don’t exclude patients with dysphagia and asthenia. Both these two conditions are very common in FM patients, but they could also satisfy the Lundberg 2017 IIM criteria. This could appear as a potential study limitation, however we believe that, on the contrary, it is an interesting point, as already discussed in introduction, line 63-69. These two symptoms could be underestimated in the context of a clinically evident FM, whereas the presence of FM doesn’t exclude the presence of another underlying condition in general and IIM in particular. This point is clearly assessed in the 2016 diagnostic criteria of FM [ref. 14] and we discussed this point in the introduction section. We clarified this part in the method section, line 96-97 and line 107-109
  3. We added the full database as supplementary material as you suggested.
  4. Unfortunately, ANA positivity was recorded with the specific pattern, however the images were not collected. The paragraph in the method section about the evaluation of autoantibodies was clarified in line 120-145.

Thank you again for your valuable comments, we hope that this new version could satisfy you. 

Round 2

Reviewer 2 Report

Thankyou for giving me the possibility to review the revised version of the paper "Myositis-Specific and Myositis-Associated Antibodies in Fibromyalgia patients: a prospective study". 

All the required changes have been made in the revised version of the manuscript. 

The paper is now suitable for publication. 

Author Response

Thank you for your comments and your time in revising our manuscript

Kind Regards

Reviewer 3 Report

The response of the authors is considered satisfactory.

Minor comments to be appreciated by readers are:

Line 86: briefly explain meaning of “circa”

The supplementary table should add a key tab to explain abbreviations used.

Author Response

Thank you for your comments and your time in revising our manuscript

Line 86 was fixed. Sorry, "circa" means "about"

We added in the supplementary materials a tab called "explanation" with all the abbreviations used

Kind Regards